

# Construction of an immunoinformatics-based multi-epitope vaccine candidate targeting Kyasanur forest disease virus

Sunitha Manjari Kasibhatla[1,*], Lekshmi Rajan[2,*], Anita Shete[2], Vinod Jani[1], Savita Yadav[2], Yash Joshi[2], Rima Sahay[2], Deepak Y. Patil[2], Sreelekshmy Mohandas[2], Triparna Majumdar[2], Uddhavesh Sonavane[1], Rajendra Joshi[1] and Pragya Yadav[2]

[1] Centre for Development of Advanced Computing, Pune, India
[2] Indian Council of Medical Research-National Institute of Virology, Pune, India
* These authors contributed equally to this work.

Corresponding authors
Rajendra Joshi,
rrjoshi02@yahoo.com
Pragya Yadav,
hellopragya22@gmail.com

## ABSTRACT

Kyasanur forest disease (KFD) is one of the neglected tick-borne viral zoonoses. KFD virus (KFDV) was initially considered endemic to the Western Ghats region of Karnataka state in India. Over the years, there have been reports of its spread to newer areas within and outside Karnataka. The absence of an effective treatment for KFD mandates the need for further research and development of novel vaccines. The present study was designed to develop a multi-epitope vaccine candidate against KFDV using immunoinformatics approaches. A total of 74 complete KFDV genome sequences were analysed for genetic recombination followed by phylogeny. Computational prediction of B- and T-cell epitopes belonging to envelope protein was performed and epitopes were prioritised based on IFN-Gamma, IL-4, IL-10 stimulation and checked for allergenicity and toxicity. The eight short-listed epitopes (three MHC-Class 1, three MHC-Class 2 and two B-cell) were then combined using various linkers to construct the vaccine candidate. Molecular docking followed by molecular simulations revealed stable interactions of the vaccine candidate with immune receptor complex namely Toll-like receptors (TLR2-TLR6). Codon optimization followed by *in-silico* cloning of the designed multi-epitope vaccine construct into the pET30b (+) expression vector was carried out. Immunoinformatics analysis of the multi-epitope vaccine candidate in the current study has potential to significantly accelerate the initial stages of vaccine development. Experimental validation of the potential multi-epitope vaccine candidate remains crucial to confirm effectiveness and safety in real-world conditions.

# INTRODUCTION

Kyasanur forest disease is a highly neglected emerging tick-borne viral zoonosis caused by Kyasanur forest disease virus (KFDV) that belongs to the family *Flaviviridae* (*Gould & Solomon, 2008*). The disease was first identified in 1957 in the Kyasanur forest region of Karnataka, India, and has since posed a significant public health threat, with sporadic

outbreaks occurring in newer districts of Karnataka and the states of Maharashtra, Goa, Kerala, and Tamil Nadu (*Yadav, Sahay & Mourya, 2018*; *Patil et al., 2017*; *Gladson et al., 2021*; *Tandale et al., 2015*). The mortality rate for KFDV infection is reported to be about 2–10% (*Gladson et al., 2021*; *National centre for disease control (NCDC), 2018*). Mammals and birds act as secondary hosts as well as reservoir for transmission of KFDV through ticks to the vertebrate hosts, with the main vector being the anthropophagic *Haemaphysalis spinigera* (*Boshell & Rajagopalan, 1968*; *Trapido et al., 1959*; *Work, 1958*).

KFDV is an enveloped spherical virus with single-stranded RNA of 11 kb size enclosed in the icosahedral nucleocapsid. The diameter of the virion is 40–65 nm and codes for a single polyprotein (*Dodd et al., 2011*; *Gritsun et al., 2014*). Three structural (capsid, pre-membrane (prM) and envelope) and seven non-structural proteins (NS1, NS2A, NS2B, NS3, NS4A, NS4B and NS5) are encoded by the polyprotein. The envelope (E) protein of KFDV plays an important role in the entry of the pathogen into the host cells by receptor binding and membrane fusion (*Mukhopadhyay, Kuhn & Rossmann, 2005*; *Mondotte et al., 2007*). The E protein has three domains, domain I (EDI), domain II (EDII), and domain III (EDIII). Among them, ED III is associated with the receptor binding and initiating the first step of viral entry. Although the exact cellular receptor for KFDV has not been conclusively identified, studies on closely related flaviviruses indicate that laminin receptors and dendritic cell-specific intercellular adhesion molecule 3-grabbing non-integrin (DC-SIGN) could be involved (*Piccini, Castilla & Damonte, 2015*; *Kuno, 2007*). After binding to the receptor, the virus is internalized into the host cell *via* clathrin-mediated endocytosis (*Chu & Ng, 2004*; *Van der Schaar et al., 2007*). The role of the E protein, especially EDIII, in facilitating efficient endocytosis is significant as it maintains a stable interaction with the host cell membrane during internalization. Within the endosome, the pH drops, which triggers a conformational rearrangement in the E protein. This change is crucial for viral entry as it exposes the fusion loop in EDII, which interacts with the endosomal membrane (*Modis et al., 2004*; *Zhang et al., 2003*). Once the viral and endosomal membranes fuse, the viral RNA genome is released into the host cell cytoplasm, initiating replication and protein synthesis (*Stiasny, Koessl & Heinz, 2003*; *Rothan & Kumar, 2019*). The EDIII domain is thought to revert to its original configuration post-fusion, allowing the E protein to retain functional integrity for further rounds of viral assembly and exit (*Zhang et al., 2003*). The E protein is a key immunogenic component that contains several epitopes recognized by B and T cells. It includes multiple B-cell epitopes capable of inducing neutralizing antibody responses. Additionally, the E protein plays a crucial role in T-cell activation, as vaccination strategies involving E protein constructs have been shown to enhance the activation of both CD4+ and CD8+ T cells which are essential mediators of cellular immunity. These T cells produce cytokines such as TNF-$\alpha$ and IFN-$\gamma$, which contribute to an amplified immune response against KFDV (*Sirmarova et al., 2018*). Furthermore, studies utilizing a vesicular stomatitis virus (VSV)-based platform expressing KFDV prM and E proteins have shown that immunized nonhuman primates develop strong neutralizing antibody titers against KFDV, as well as cross-reactive antibodies against Alkhurma hemorrhagic fever virus (AHFV), suggesting potential cross-protective benefits within the flavivirus group (*Bhatia et al., 2023*; *Bhatia*

*et al., 2021*). E-protein evokes the neutralizing antibody response, which has been demonstrated to effectively neutralize KFDV and related flaviviruses, making it a valuable target for vaccine development (*Heinz & Stiasny, 2012*).

Presently, there is no clinically approved drug against KFDV and hence affected individuals are treated based on symptoms along with supportive therapy. The formalin-inactivated KFD vaccine was one of the early measures used in preventing the disease and has been used primarily in certain regions of India where KFD is endemic. However, the primary concern with the formalin-inactivated vaccine is its hesitancy by people due to side effects at the injection site (*Dandawate et al., 1994*; *Kasabi et al., 2013b*). While it may offer some protection against KFDV, it may not provide complete or long-lasting immunity. There may be breakthrough infections among individuals who have received the vaccine. Earlier studies demonstrated the low effectiveness of the vaccine even after repeated booster doses (*Kasabi et al., 2013a*; *Kiran et al., 2015*). Besides this, the estimated mean rate of nucleotide substitution based on the complete genome of KFDV is $4.2 \times 10^{-4}$ subs/site/year (*Yadav et al., 2020*). The earliest known KFDV strain was isolated in 1957, so the virus has been in circulation since the last six decades in India. Studies have shown that KFDV can accumulate mutations in its envelope protein. Some of these mutations have been linked to changes in the antigenicity of the envelope protein, which means that the virus may become less susceptible to the immune system (*Yadav et al., 2020*; *Mehla et al., 2009*; *Shil et al., 2018*). The low efficacy of currently available vaccines and the increasing spread of KFDV to naive regions in India necessitates the need for the development of a new KFD vaccine (*Yadav, Sahay & Mourya, 2018*; *Kasabi et al., 2013a*; *Kiran et al., 2015*).

Newer vaccine development techniques, such as recombinant DNA technology or viral vector vaccines, may offer advantages in terms of efficacy and safety. Computational vaccinology and *in-silico* prediction of the host's immune response to pathogens/antigens can expedite the development of novel vaccines. Epitope-based vaccines, which target specific antigenic regions of the pathogen, offer a promising approach to vaccine development termed 'reverse vaccinology' due to their potential for inducing targeted immune responses while minimizing side effects (*Rappuoli, 2000*). Computational epitope prediction and immuno-informatics can hasten the design and development of multi-epitope vaccines (*Pizza et al., 2000*; *Masignani, Pizza & Moxon, 2019*). Molecular docking and molecular simulations aid in understanding the binding potential of *in-silico* designed vaccine with host immune cell receptors (*Rappuoli et al., 2016*). Immunoinformatics has been gainfully employed for designing multi-epitope vaccine candidates against KFDV (with a relatively smaller set of genome sequences) and to understand the molecular interactions between the host receptors and virus protein recently (*Arumugam & Varamballi, 2021*; *Dey et al., 2023*; *Hafeez et al., 2023*). The present work is a comprehensive phylogenomic and immunoinformatics analysis of KFDV based on 73 genome sequences to rationally design a multi-epitope vaccine candidate based on a major antigenic protein, which is an envelope protein. Portions of this text were previously published as part of a preprint (https://www.biorxiv.org/content/10.1101/2024.03.14. 584963v1).

## MATERIALS AND METHODS

### Study design

The complete genome sequences of the KFDV isolates available with ICMR-National Institute of Virology, Pune, India were analyzed using various bioinformatics tools to determine the lineages and the level of genetic recombination among the isolates. The study aims to design a vaccine candidate that can induce a strong and specific immune response by targeting the envelope protein that plays a key role in the virus-host interaction. However, it is also important to consider the diversity of KFDV strains and the potential for antigenic variation when designing vaccines based on the envelope protein to ensure broad protection. Considering this, various immunoinformatics tools were used in the current study to develop an *in-silico*-designed multi-epitope (B- and T-cell derived) vaccine directed towards the envelope protein of KFDV. The physicochemical properties of the vaccine construct were computed followed by prediction and refinement of its three-dimensional (3D) structure. To analyze the binding affinity and stability, molecular docking analysis and molecular dynamics simulations were performed. Figure 1 shows the different steps for designing the multi-epitope vaccine construct against KFD in our study.

### Next-generation sequencing and phylogenetic analysis

A total of 73 complete genome sequences of KFDV from India were used in this study. Out of these 49 were previously submitted to GenBank by the Indian Council for Medical Research-National Institute of Virology (ICMR-NIV) Pune, and 24 were collected during 2019–2022 and sequenced by the Next generation sequencing approach (Table S1). For next generation sequencing, RNA was extracted from isolates and clinical specimens with a commercially available RNA extraction kit (MagMAX$^{TM}$ Viral/Pathogen; Themo Fisher Scientific, Waltham, MA, USA). The libraries were prepared, pooled, and loaded onto the Illumina platform as per the protocol described earlier (*Yadav et al., 2018*). CLC Genomics software 22.0.2 (Qiagen, Germantown, MD, USA) was used for the sequence analysis. Reference mapping was carried out with the KFDV reference genome (Ref Seq ID: NC_039218) to retrieve the complete genome. Multiple sequence alignment of the KFDV isolates including the reference genome was performed using the tool Multiple Alignment using Fast Fourier Transform (MAFFT) (*Katoh, Rozewicki & Yamada, 2019*). Multiple alignments of individual genes/proteins were also carried out using MAFFT. Molecular Evolutionary Genetic Analysis version 10 (MEGAX) was used for the visualization of genome/gene/protein alignments (*Kumar et al., 2018*). Recombination events were detected using Recombination Detection Program version 4 (RDP4) with *p*-value ≤ 0.005. A sequence is said to be recombinant if predicted by at least three different methods available in RDP4 package (*Martin et al., 2015*). The best nucleotide substitution model was estimated by ModelFinder (*Kalyaanamoorthy et al., 2017*). For the construction of the phylogenetic tree, the maximum likelihood method implemented in Important Quartet based phylogenetic tree (IQTREE) (*Minh et al., 2020*) tool was used. Phylogenetic tree was visualized using Interactive Tree of Life (iTOL) (*Letunic & Bork, 2021*). Molecular clock behavior was tested using TEMPoral Exploration of Sequences and Trees (TempEst)
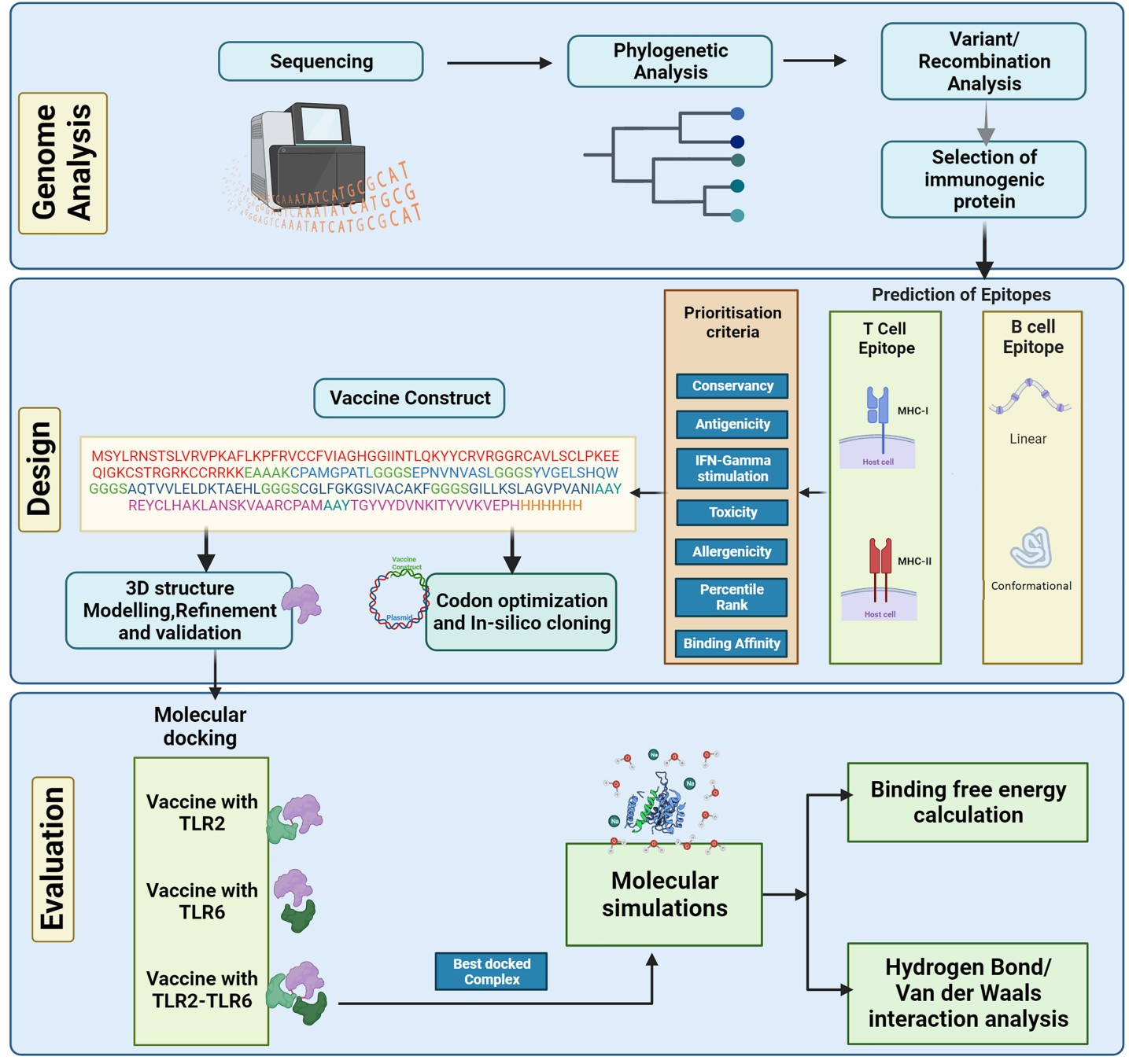

**Figure 1 Schematic representation of steps involved in the *in-silico* design of Kyasanur forest disease vaccine.**

(*Rambaut et al., 2016*). Bayesian Evolutionary Analysis Sampling Trees (BEAST) v.10.1.4 was used for the estimation of nucleotide substitution rate and lineage divergence (*Suchard et al., 2018*). One billion steps of Markov Chain Monte Carlo (MCMC) were carried out in triplicate. The maximum clade credibility tree was visualized using FigTree (available for download at http://tree.bio.ed.ac.uk/software/figtree/).

## Immunoinformatics analysis

### T cell epitope prediction

NetMHCpan EL 4.1 (available at http://tools.iedb.org/mhci/) and NetMHCIIpan 3.2 (available at http://tools.iedb.org/mhcii/) from the IEDB server were used respectively for predicting MHC I and MHC II epitopes (*Reynisson et al., 2020*) with HLA alleles as curated by *Weiskopf et al. (2013)* and *Greenbaum et al. (2011)*. Further filtration of the predicted epitopes was based on MHC binding affinity, percentile rank and by removing largely overlapping epitopes that share the same core region. Computational prediction of IL-4 and IL-10 stimulation has been carried out using IL4Pred (*Dhanda et al., 2013*) and IL-10Pred (*Nagpal et al., 2017*).

### B-cell epitope prediction

Prediction of conformational and linear B-cell epitopes was done using Bepipred (available at http://tools.immuneepitope.org/bcell/) and ElliPro (available at http://tools.iedb.org/ellipro/) online tools (*Jespersen et al., 2017*; *Ponomarenko et al., 2008*). For the conservancy of the epitopes in the global isolates, the Epitope Conservancy Analysis tool (available at http://tools.iedb.org/conservancy/) was used (*Bui et al., 2007*). Computational prediction of IFN-gamma stimulation of the predicted epitopes was performed using IFNepitope (*Dhanda, Vir & Raghava, 2013*) (available at https://webs.iiitd.edu.in/raghava/ifnepitope/predict.php). Further prioritization of the predicted epitopes was carried out by testing for antigenicity using Vaxijen 2.0 (*Doytchinova & Flower, 2007*) (available at http://www.ddg-pharmfac.net/vaxijen/). Allergenicity and toxicity testing were done using AllerTopv2.0 (available at https://www.ddg-pharmfac.net/allertop_test/) and ToxinPred (*Dimitrov et al., 2014*; *Gupta et al., 2015*) (available at https://webs.iiitd.edu.in/raghava/toxinpred/), respectively.

### Construction of multi-epitope vaccine

The selected T-cell and B-cell epitopes were conjugated with different linkers. The linker regions used were EAAAK, GGGS, and AAY. The N-terminal of the vaccine construct was linked with an adjuvant β-defensin by the EAAAK linker. MHC I and MHC II epitopes were linked by GGGS and the B cell epitopes by AAY. A histidine tag was included at the C terminal of vaccine construct which may be useful during purification steps in experimental validation studies. The use of adjuvant β-defensin would help to enhance the vaccine immunogenicity (*Kim et al., 2018*).

### Physiochemical properties

Various physiochemical properties of the vaccine construct were calculated using the online web server ProtParam (*Gasteiger et al., 2005*) (available at https://web.expasy.org/protparam).

### Prediction of 3D structure of vaccine construct

The three-dimensional structure of the vaccine construct was predicted using the I-TASSER webserver (*Zhou et al., 2022*). I-TASSER stands for Iterative Threading Assembly Refinement. It develops multiple models, and based on the C-score, the best

model is chosen. The chosen model of the vaccine construct was further refined by molecular dynamics simulations using amberff14SB force field available in AMBER20 simulation package (*Case et al., 2021*). The system preparation was done where the structure was solvated in a water box using the TIP3P model. The system was neutralized by adding charged (18 chloride) ions. The structure was minimized in two stages, namely steepest descent followed by conjugate gradient approaches. Minimization was followed by temperature ramping up to 300 K since the amberff14SB force field (used in this study) with TIP3P water model is parameterized at 300 K (*Maier et al., 2015*). This was followed by an equilibration at NPT condition for one ns, and finally, a production run of 50 ns was carried out. The final model obtained from the simulations was validated by Ramachandran plot analysis using the Procheck webserver (*Laskowski et al., 1993*) (available at https://www.ebi.ac.uk/thornton-srv/software/PROCHECK/).

## Molecular docking with immune cell receptors

The vaccine construct must interact with the host immune cell receptors like Toll-like receptors (TLR) to evoke an immune response. Hence, the binding affinity of TLR2, TLR6 and TLR2-TLR6 receptor complex with the vaccine construct were studied using molecular docking. HADDOCK2.4 webserver (*Dominguez, Boelens & Bonvin, 2003*) (available at https://rascar.science.uu.nl/haddock2.4/) was used for docking studies with TLR2 receptor (PDB ID: 2z7x), TLR6 (PDB ID: 3a79), and TLR2-TLR6 receptor complex (PDB ID: 3a79) (*Dominguez, Boelens & Bonvin, 2003*; *van Zundert et al., 2016*). Chimera software (*Pettersen et al., 2004*) was used for visualization and for removing the hetero water molecules and pam2SCK4 from PDB files. Prior to docking, the active and passive residues were predicted using WHISCY (*De Vries, van Dijk & Bonvin, 2006*) (available at https://wenmr.science.uu.nl/whiscy/). HADDOCK provides multiple docked cluster models. Based on the lowest HADDOCK score, the best-docked cluster was chosen and subjected to molecular dynamics simulations.

## Molecular simulations of TLR2-TLR6-vaccine construct complex

Molecular dynamics simulations were performed for the docked TLR2-TLR6-vaccine construct complex using the AMBER20 simulation package (*Case et al., 2021*). A similar protocol to that of the vaccine construct was followed for the simulations, except that an additional production run of 240 ns was performed. A total of three sets of simulations were performed. Interaction analysis and related statistical analysis were performed for the last 200 ns of simulation data. Root mean square deviation (RMSD), root mean square fluctuation (RMSF), and hydrogen bond analysis were performed using AmberTools21 (*Case et al., 2021*). RMSD indicates how much the structure has deviated from the reference structure. The reference structure considered for the RMSD calculation is the start structure of the simulation. Interaction calculations were performed using the GetContact tool (available at https://getcontacts.github.io/) and the LIGPLOT tool (*Laskowski & Swindells, 2011*). Free energy of binding between the vaccine construct and TLR2-TLR6 receptor, vaccine construct, and TLR2 receptor, and vaccine construct and

TLR6 receptor were computed using the MM-GBSA module of the AMBER20 package. Secondary structure prediction was carried out using PSIPRED4.0 (*Buchan & Jones, 2019*).

## Codon optimization and *in-silico* cloning

Reverse translation and codon optimization was performed using VectorBuilder (https://en.vectorbuilder.com/tool/codon-optimization.html) to achieve a high protein expression level in *E.coli* (strain K12) for future validation studies. The codon-optimized sequence was used for *in-silico* cloning into pET30b (+) using XhoI and NdeI (https://www.snapgene.com/).

# RESULTS

## Analysis of KFDV genomes

There was no evidence of recombination in the KFDV isolates. A linear relationship between root-to-tip distance and time of isolation was observed indicating molecular clock behavior in the dataset (Fig. S1).

## Phylogenetic analysis

Genome-wide nucleotide substitution rate was found to be $4.31 \times 10^{-4}$ (95% HPD: $3.45 \times 10^{-4}$, $5.22 \times 10^{-4}$) substitutions per site per year. Phylogenetic analysis revealed spatio-temporal clustering with few deviations (Fig. 2). Two distinct lineages of KFDV were found, of which, lineage 1 includes isolates sampled from Karnataka during 1957–1972. Lineage 2 was found to demarcate into sub-lineages viz., 2.1 and 2.2. Sub-lineage 2.1 includes isolates from Karnataka and Goa during 2006–2022. Sub-lineage 2.2 further differentiates into 2.2.1 and 2.2.2. It was observed that ten KFDV isolates from Karnataka sampled during 2012–2022 constitute lineage 2.2.1. KFDV strains isolated during 2013–2020 from Maharashtra along with six from Karnataka, two from Goa, and one from Tamil Nadu constitute sub-lineage 2.2.2. Nine representative isolates were identified based on the lineages observed in the phylogenetic tree (Table S2).

## T-cell epitope prediction

MHC class I epitope prediction for envelope protein was carried out for reference HLA alleles as curated by *Weiskopf et al. (2013)*. The predicted epitopes were further filtered based on percentile rank ≤ 1 and predicted binding affinity of 500 nM which resulted in a total of 23 MHC class I epitopes (Table 1). Similarly, MHC class II epitopes were predicted for the HLA reference set curated by *Greenbaum et al. (2011)*. The class II epitopes were prioritized based on removing largely overlapping epitopes that share the same core region, MHC binding affinity IC50 ≤ 1,000 nM, and percentile rank ≤ 10.0 which resulted in a total of 15 MHC class II epitopes (Table 2). Among the short-listed MHC Class II epitopes except epitope 262-GILLKSLAGVPVANI-276, all epitopes were positive for stimulation of both IL-4 and IL-10 (Table S3).

## B-cell epitope prediction

Linear B-cell epitopes were predicted using the envelope protein sequences of the nine representative KFDV isolates which were further prioritized based on their conservancy

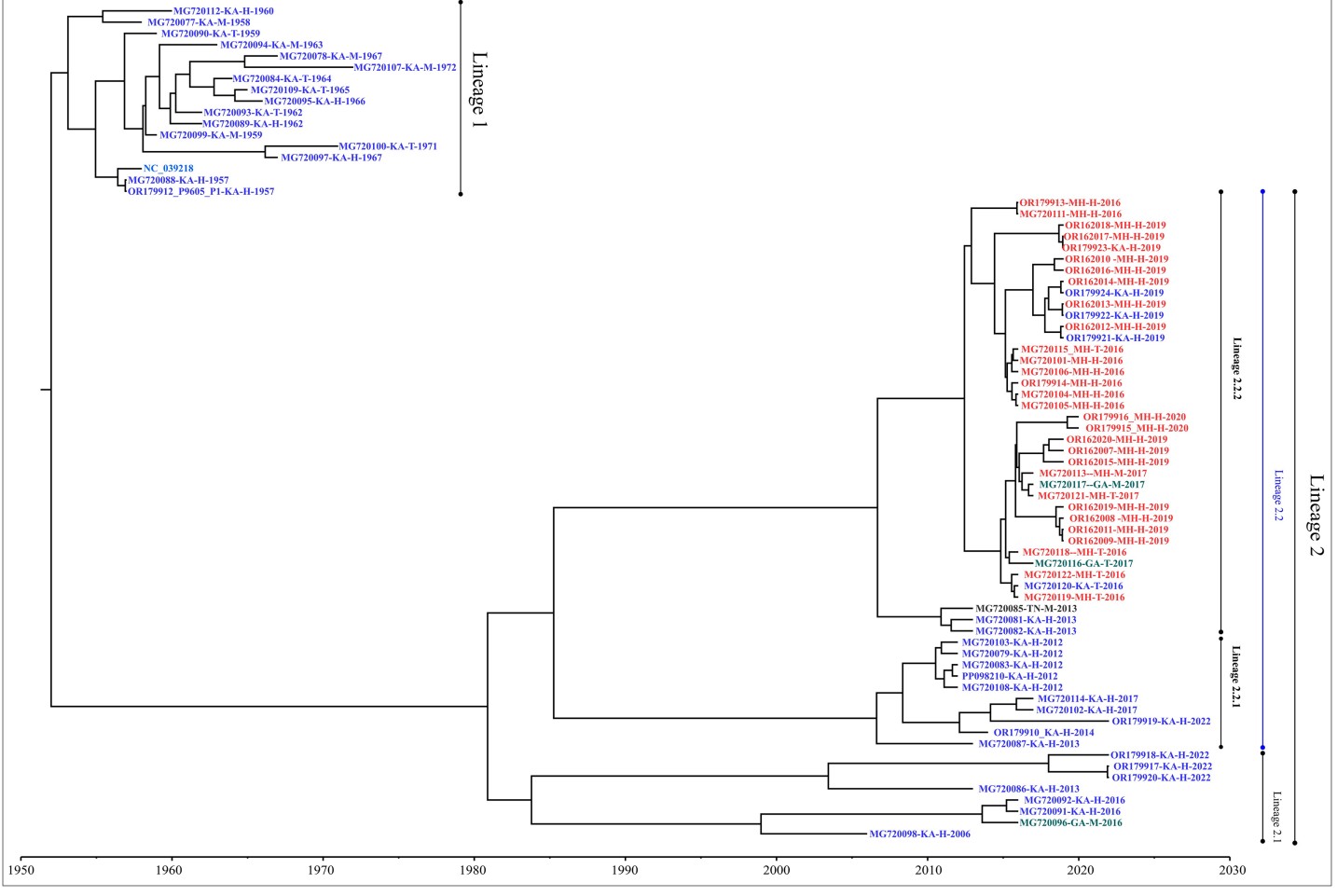

**Figure 2 Maximum clade credibility tree of KFDV derived using complete genome.**

across all global KFDV isolates, the ability to induce IFN-Gamma stimulation, and antigenicity (Table 3). The predicted 3D structure of envelope protein was used for conformational epitope prediction and three epitopes were prioritized based on similar criteria as described above for linear B-cell epitopes (Table 4).

## Construction of multi-epitope vaccine candidate

The prioritized epitopes were combined with the help of different linkers, β-defensin adjuvant, and histidine tag. The length of the final vaccine construct is 226 amino acids with a molecular weight of 23.09 kDa (Fig. 3). The vaccine construct has a theoretical pI value of 9.54, suggesting its alkaline nature and the estimated half-life is 30 h (mammalian reticulocytes, *in-vitro*). The computed instability index (II) was found to be 39.08 which classifies the protein as stable. The average hydropathicity was around −0.029, indicating the hydrophilic nature of the vaccine construct.

**Table 1 List of short-listed MHC-Class I epitopes belonging to the envelope protein of KFDV.**

| Epitope | Position | IFN-Gamma | Antigenicity | Percent of protein sequence matches at identity <= 100% | Rank | IC50 |
|---|---|---|---|---|---|---|
| STIGRVLEK | 397–405 | Positive | Non-antigen | 100.00% (74/74) | 0.01 | 21,121.52 |
| KAWQVHRDW | 211–219 | Positive | Non-antigen | 100.00% (74/74) | 0.01 | 9,711.53 |
| LTVVGEHAW | 413–421 | Negative | Antigen | 98.65% (73/74) | 0.01 | 19,183.08 |
| YVGELSHQW | 384–392 | Positive | Antigen | 98.65% (73/74) | 0.01 | 69.87 |
| FLPRILLGV | 454–462 | Negative | Non-antigen | 100.00% (74/74) | 0.04 | 14,056.58 |
| ILLKSLAGV | 263–271 | Positive | Non-antigen | 100.00% (74/74) | 0.05 | 21,522.7 |
| TRASLVLEL | 19–27 | Positive | Antigen | 2.70% (2/74) | 0.01 | 21,522.7 |
| KLKMKGMTY | 296–304 | Negative | Antigen | 100.00% (74/74) | 0.01 | 15,983.96 |
| ASFTTQSEK | 163–171 | Negative | Antigen | 100.00% (74/74) | 0.01 | 17,634.96 |
| LPPGDNIIY | 376–384 | Negative | Non-antigen | 100.00% (74/74) | 0.04 | 17,158.59 |
| QEWNHANRL | 233–241 | Negative | Antigen | 48.65% (36/74) | 0.04 | 24,903.62 |
| TRVSLVLEL | 19–27 | Positive | Antigen | 97.30% (72/74) | 0.01 | 19,937.05 |
| VYDVNKITY | 131–139 | Negative | Non-antigen | 100.00% (74/74) | 0.01 | 11,777.84 |
| NHADRLVEF | 236–244 | Negative | Non-antigen | 51.35% (38/74) | 0.01 | 1,969.83 |
| CPAMGPATL | 74–82 | Positive | Antigen | 100.00% (74/74) | 0.03 | 27.6 |
| VEFGEPHAV | 242–250 | Positive | Antigen | 100.00% (74/74) | 0.01 | 27,602.87 |
| EPNVNVASL | 349–357 | Positive | Antigen | 100.00% (74/74) | 0.03 | 256.83 |
| EHLPKAWQV | 207–215 | Negative | Non-antigen | 100.00% (74/74) | 0.02 | 25,442.61 |
| KMKGMTYTV | 298–306 | Negative | Non-antigen | 100.00% (74/74) | 0.02 | 7,773.82 |
| VANIEGSKY | 273–281 | Negative | Non-antigen | 95.95% (71/74) | 0.05 | 7,450.51 |
| NHANRLVEF | 236–244 | Negative | Non-antigen | 48.65% (36/74) | 0.02 | 19,651.14 |
| LQLPPGDNI | 374–382 | Negative | Non-antigen | 100.00% (74/74) | 0.03 | 26,164.55 |
| FGEPHAVKM | 244–252 | Negative | Non-antigen | 100.00% (74/74) | 0.02 | 19,680.08 |

**Note:**
Epitopes used in the design of vaccine construct are underlined.

## Prediction of 3D structure, refinement, and validation

For the prediction of tertiary structure, the I-TASSER server was used. The server predicted five models, and based on the C-score, the top-scoring model was selected for further refinement. I-TASSER generates three dimensional atomic structure based on the multiple threading alignments and iterative structural assembly simulations. The threading templates PDB IDs (%identity) used were 3p54 (19%), 7cth (16%), 7esd (20%), 2i69 (23%), 6epk (20%), 4fg0 (22%), 7w6b (8%) and 1svb (20%). The structure was refined with the help of molecular dynamics simulations. The structural quality of the predicted and refined vaccine 3D structure was validated by the Ramachandran plot (Fig. S2). ~98.4% of amino acids belong to the most favored, additional allowed, and generously allowed regions. Three residues, *i.e.*, 1.6%, belonged to the disallowed region. Overall, the model was observed to have good quality with predicted Z score of −3.54 using ProSA analysis (*Sippl, 1993*).

**Table 2 List of short-listed MHC-Class II epitopes belonging to the envelope protein of KFDV.**

| Epitope | Position | IFN-Gamma | Antigenicity | Percent of protein sequence matches at identity <= 100% | Rank | ic50 |
|---|---|---|---|---|---|---|
| AQTVVLELDKTAEHL | 195–209 | Positive | Antigen | 98.65% (73/74) | 9.3 | 30.7 |
| AVAHGEPNVNVASLI | 344–358 | Positive | Antigen | 100.00% (74/74) | 9.9 | 98.5 |
| CGLFGKGSIVACAKF | 105–119 | Positive | Antigen | 100.00% (74/74) | 1.6 | 25.3 |
| DQTGILLKSLAGVPV | 259–273 | Positive | Antigen | 100.00% (74/74) | 1.21 | 6.9 |
| EFGEPHAVKMDIFNL | 243–257 | Negative | Non-antigen | 100.00% (74/74) | 7.5 | 123 |
| EGKPSVDVWLDDIHQ | 36–50 | Positive | Antigen | 100.00% (74/74) | 2.8 | 280.1 |
| EKTILTLGDYGDISL | 170–184 | Negative | Non-antigen | 94.59% (70/74) | 6.4 | 357.7 |
| GHDTVVMEVTYTGSK | 322–336 | Positive | Antigen | 100.00% (74/74) | 8.6 | 197.5 |
| GIERLTVVGEHAWDF | 409–423 | Negative | Non-antigen | 98.65% (73/74) | 7.8 | 726.2 |
| GILLKSLAGVPVANI | 262–276 | Positive | Antigen | 100.00% (74/74) | 0.38 | 3.4 |
| LPKAWQVHRDWFEDL | 209–223 | Positive | Antigen | 100.00% (74/74) | 7.4 | 101.5 |
| PGDNIIYVGELSHQW | 378–392 | Negative | Non-antigen | 98.65% (73/74) | 8.8 | 312.5 |
| REYCLHAKLANSKVA | 57–71 | Positive | Antigen | 100.00% (74/74) | 2.8 | 15.8 |
| RKTASFTTQSEKTIL | 160–174 | Positive | Antigen | 100.00% (74/74) | 9.5 | 849.5 |
| SGTQGTTRASLVLEL | 13–27 | Negative | Non-antigen | 2.70% (2/74) | 8.2 | 38.6 |

Note:
Epitopes used in the vaccine construct are underlined.

**Table 3 List of predicted linear B-cell epitopes belonging to the envelope protein of KFDV.**

| Epitope sequence | Epitope length | Position | IFN-Gamma | Antigenicity | Percent of protein sequence matches at identity <= 100% |
|---|---|---|---|---|---|
| REYCLHAKLANSKVAARCPAM | 21 | 57–77* | Yes | Yes | 100.00% (74/74) |
| EHLPKAWQVHRD | 12 | 206–218* | No | No | 100.00% (74/74) |
| GILLKSLAGVPVANI | 15 | 262–276 | No | No | 100.00% (74/74) |
| SKPCRIPVRAVAHGE | 15 | 334–349 | No | Yes | 100.00% (74/74) |
| DISLTCRVTSGVDPAQTVVLELD | 23 | 180–203 | No | Yes | 97.30% (72/74) |
| FGGVGFLPRILLGVALAWLG | 20 | 448–468* | No | Yes | 95.95% (71/74) |
| TGYVYDVNKITYVVKVEPH | 19 | 127–146 | Yes | Yes | 100.00% (74/74) |
| SKYHLQSGHVTCDVGLE | 17 | 279–295 | No | Yes | 95.95% (71/74) |
| GGMLSSVGKALHTAFGA | 17 | 425–443 | Yes | Yes | 81.08% (60/74) |
| RVSLVLELGGCVTLT | 15 | 19–34 | No | Yes | 97.30% (72/74) |
| KGSIVACAKFSCE | 13 | 108–122* | No | Yes | 95.95% (71/74) |
| NIIYVGELSHQW | 12 | 380–392 | No | Yes | 98.65% (73/74) |

Note:
Epitopes short-listed to design vaccine construct are underlined. Asterisk indicates experimentally validated epitopes in other *Flavivirus* members (*Fumagalli, Figueiredo & Aquino, 2021*).

## Docking of vaccine construct with host receptors (TLR2, TLR6, and TLR2-TLR6 complex)

Docking of vaccine construct was performed with individual TLRs (TLR2 and TLR6) as well as TLR2-TLR6 complex. Table 5 shows the statistics for the refined model with TLR2,

**Table 4 List of predicted conformational B-cell epitopes belonging to the envelope protein of KFDV.**

| Epitope | Epitope length | Position | IFN-Gamma | Antigenicity | % of protein sequence matches at identity <= 100% | Score |
|---|---|---|---|---|---|---|
| LEKLKMKGMTYTVCEGSKFAWKRPPT | 26 | 294–319 | Yes | No | 100.00% (74/74) | 0.704 |
| LPPGDNIIYVGELSHQWFQKGSTIG | 25 | 376–400 | No | Yes | 100.00% (74/74) | 0.7 |
| PSMETTGGG | 9 | 362–370 | No | Yes | 98.65% (73/74) | 0.685 |
| EGAQEW | 6 | 230–235 | Yes | No | 98.65% (73/74) | 0.583 |
| TTQSEKTILTLGDYGD | 16 | 166–181 | No | Yes | 100.00% (74/74) | 0.568 |
| VGGMLSSVGKALHTAFGAA | 19 | 426–444 | No | Yes | 100.00% (74/74) | 0.644 |
| LPRILLGVALAWLGLNSRNPTLSVGFLITGGLVL | 34 | 455–488 | No | Yes | 100.00% (74/74) | 0.845 |
| HAKLANSKVAARCPA MGPATLPEEHQASTV CRRDQSDRGWGNHC GLFGKGSIVACAKFSCETK | 63 | 62–124 | Yes | Yes | 100.00% (74/74) | 0.81 |
| EFGEPHAVKMDIFNL | 15 | 243–257 | No | No | 100.00% (74/74) | 0.767 |
| MEVTYTGSKPCRIPVRAVAHGEPNVNVA | 28 | 328–355 | No | Yes | 100.00% (74/74) | 0.709 |
| TQGTTR | 6 | 15–20 | No | No | 100.00% (74/74) | 0.565 |

Note:
Epitopes short-listed to design vaccine-construct are underlined.

TLR6, and TLR2-TLR6 complex. As TLR2 is known to form heterodimer with TLR6, docking with this complex was further investigated. Docking of TLR2-TLR6 complex with vaccine construct (Figs. 3B and 3C) resulted in 73 structures that group into 10 cluster(s). These represent 36.5% of the generated water-refined models. The top cluster was considered the most reliable based on the HADDOCK score. Further refinement of a representative model of the top cluster resulted in nine structures that are 100% water-refined models. The proximity between the vaccine construct and TLR receptors is indicated by a buried surface area of 4,564.2 +/− 299.3 Å$^2$. The RMSD of the docked complex is around 0.6 +/− 0.2, further reiterating the model's good quality (*Poli, Martinelli & Tuccinardi, 2016*; *Boittier et al., 2020*; *Collins et al., 2024*).

## Molecular simulations of TLR2-TLR6-vaccine construct

The stability of the binding between TLR2-TLR6-vaccine construct was assessed by calculating distances between amino acid residues belonging to vaccine construct, TLR2 and TLR6 (Figs. S3A and S3B). The figure clearly depicts that the distance is maintained for most of the simulation time indicating stable binding of vaccine construct with TLR2-TLR6. RMSD for the entire complex and individual proteins, *i.e.*, TLR2, TLR6 receptors, and vaccine construct, were computed (Fig. 4). The RMSD value for the entire protein lies in the range of 4 to 5 Å. For the TLR2 and TLR6 receptors, it is in the range of 2 to 3 Å, and for the vaccine construct, it is in the range of 6 to 7 Å. This clearly shows that the RMSD values tend to stabilize over time, indicating equilibration of the system. The higher RMSD value in vaccine construct may be attributed to the loop regions. Further, to inspect which regions of the vaccine construct show higher fluctuations, RMSF was calculated for the vaccine construct (Fig. 5). From Fig. 5, it can be observed that the region which lie in

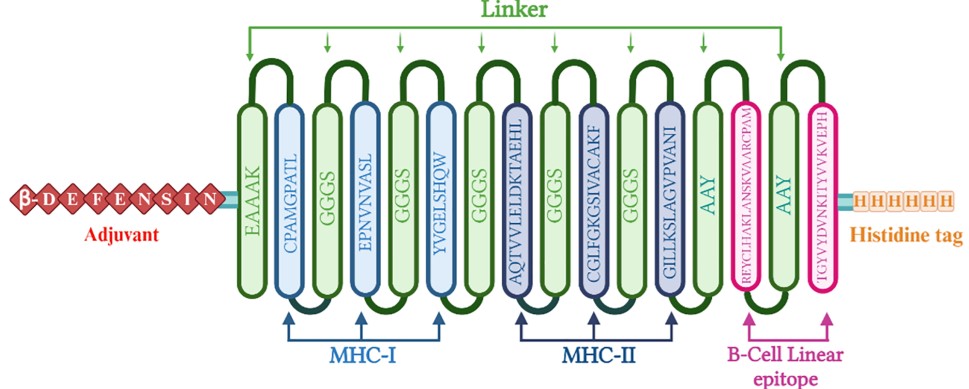

(A)

MSYLRNSTSLVRVPKAFLKPFRVCCFVIAGHGGIINTLQKYYCRVRGGRCAVLSCLPKEE
QIGKCSTRGRKCCRRKKEAAAKCPAMGPATLGGGSEPNVNVASLGGGSYVGELSHQW
GGGSAQTVVLELDKTAEHLGGGSCGLFGKGSIVACAKFGGGSGILLKSLAGVPVANIAAY
REYCLHAKLANSKVAARCPAMAAYTGYVYDVNKITYVVKVEPHHHHHHH

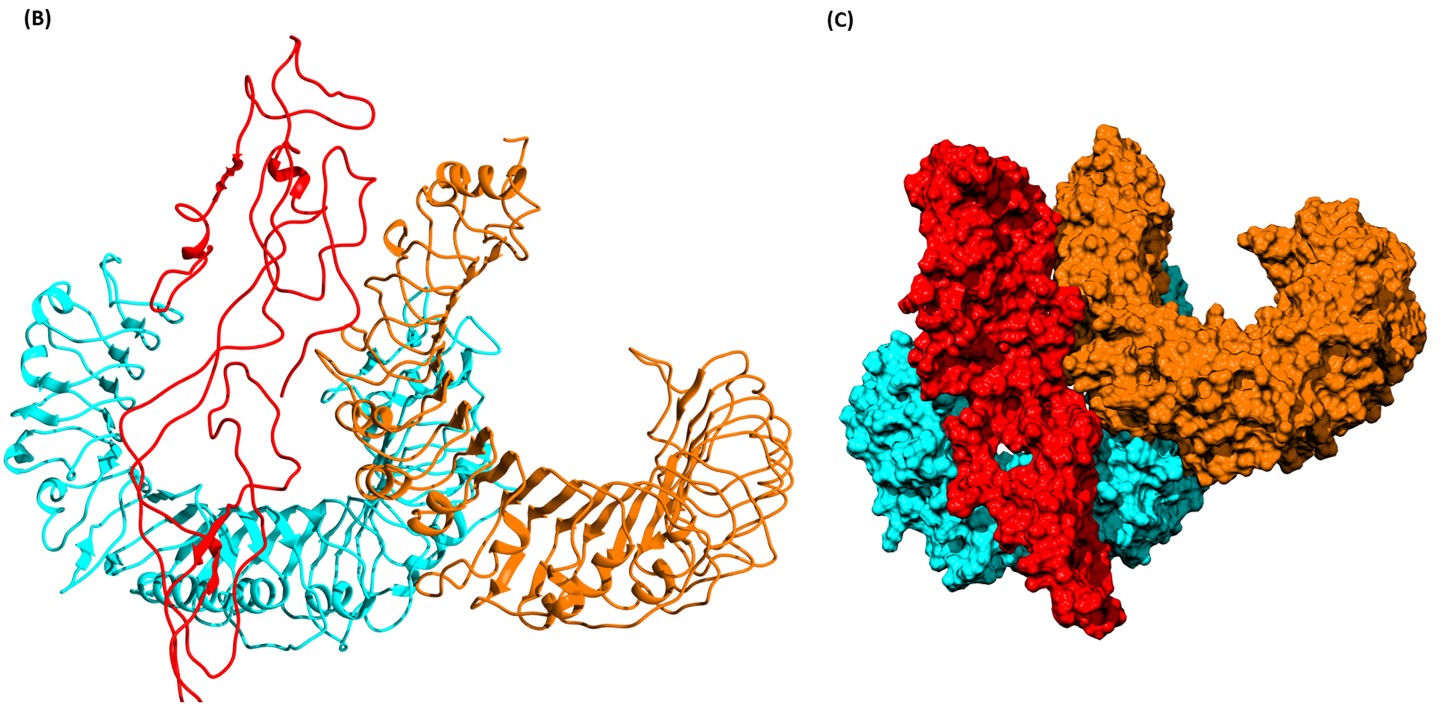

**Figure 3** **(A) Schematic representation of multi-epitope vaccine with adjuvant at the N-terminal and His-tag at the C-terminal.** (B) Three dimentional structure of the docked complex comprising KFDV vaccine construct with human receptor complex of TLR2-TLR6 shown in ribbon model. (C) Three dimentional structure of the docked complex shown as space-filled model. Color code: Vaccine construct is depicted in red, TLR6 is in blue, and TLR2 is shown in orange.

**Table 5 Statistical parameters for interaction between vaccine construct with TLR2, TLR6, and TLR2-TLR6 complex.**

| Parameter | TLR2 | TLR6 | TLR2-TLR6 |
|---|---|---|---|
| HADDOCK score | −20.2 +/− 3.8 | −82.8 +/− 37.2 | −13.0 +/− 9.3 |
| Cluster size | 19 | 9 | 9 |
| RMSD from the overall lowest-energy structure | 0.8 +/− 0.1 | 1.4 +/− 1.0 | 0.6 +/− 0.2 |
| Van der Waal's energy (kcal/mol) | −124.0 +/− 19.1 | −146.2 +/− 11.1 | −117.8 +/− 14.8 |
| Electrostatic energy (kcal/mol) | −302.0 +/− 49.0 | −501.1 +/− 63.6 | −526.7+/− 32.8 |
| Desolvation energy (kcal/mol) | −36.8 +/− 6.2 | −10.9 +/− 3.8 | −9.2 +/− 6.2 |
| Restraints violation energy (kcal/mol) | 2,009.5 +/− 219.22 | 1,745.3 +/− 162.66 | 6,996.5 +/− 225.43 |
| Buried surface area | 3,996.0 +/− 365.4 | 4,874.8 +/− 323.6 | 4,564.2 +/− 299.3 |
| Z-Score | −2.3 | −2.1 | −2 |

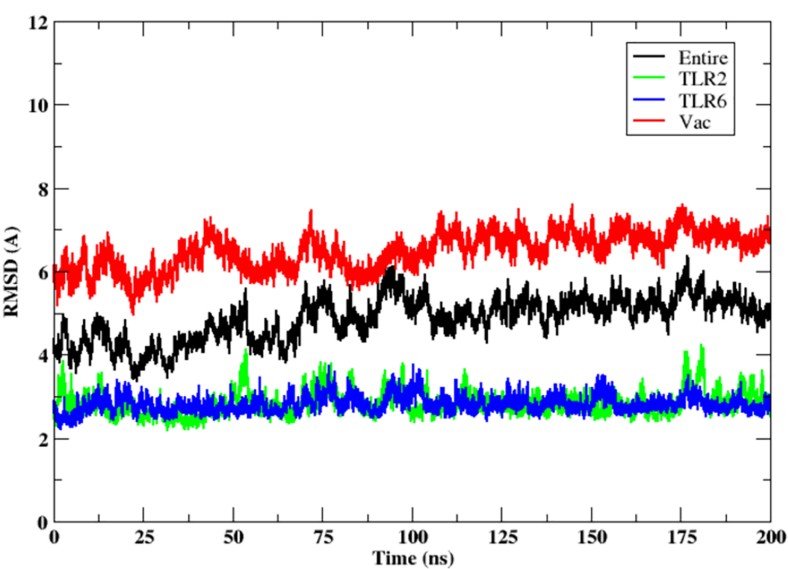

**Figure 4 RMSD of the 3D structure of vaccine construct with TLR2-TLR6 docked complex. RMSD of the entire complex (black), vaccine construct (red), TLR2 receptor (green) and TLR6 receptor (blue).**

residue range of 80–100 and 130–150 show higher fluctuations. Two replicates (two and three) show similar trends for RMSD and RMSF distribution (Figs. S4 and S5).

GetContact tool was used for the calculation of hydrogen bonds and Van der Waal's interactions between vaccine construct and TLR2–TLR6. All the hydrogen bonds and Van der Waal's interactions having occupancy of ≥20% have been shown in Tables S4, S5 and Figs. S6, S7. TLR2 forms eleven hydrogen bonds with the vaccine construct. Among these, four hydrogen bonds are between the epitope regions and TLR2, and the remaining seven hydrogen bonds are between the linker region or β-defensin region and TLR2. Besides hydrogen bonds, the vaccine construct formed thirty-eight Van der Waal's interactions with TLR2. Of these, twenty-six interactions were between epitope regions of the vaccine construct and TLR2. Similarly, TLR6 formed twenty-six hydrogen bonds and sixty-two Van der Waal's interactions with the vaccine construct. Of the twenty-six hydrogen bonds,

seven were between epitope regions and TLR6; fourteen were between the adjuvant and the TLR6. Similarly, of the sixty-two Van der Waal's interactions, twenty-five involved epitope regions. Thus, the TLR6 receptor has maximum number of interactions with vaccine construct as compared to TLR2.

Further, the free energy of binding between different components of the complex, namely, vaccine construct-TLR2/TLR6, vaccine construct and TLR2-TLR6 complex, and between TLR2-TLR6, were computed and are shown in Fig. 6. It can be observed that the vaccine construct shows strong free energy of binding with TLR2-TLR6 complex with values around −80 to −100 kcal/mol. Among the two TLRs, the vaccine construct showed better binding free energy with TLR6 as compared to TLR2. The vaccine construct and TLR2 binding free energy lies in the range of −25 to −27 kcal/mol, while it lies in the range of −60 to −75 kcal/mol for the vaccine construct and TLR6. The free energy of binding between two TLRs lies in the range of 0 to 5 kcal/mol, which indicates weak binding. Replicate two and replicate three show similar trends for free energy of binding (Fig. S8).

### Codon optimization and *in-silico* cloning

Vector Builder was used for codon optimization in the *Escherichia coli* (strain K12) expression system. The GC content was 59.43% and Codon Adaptation Index (CAI) was 0.95. A codon-optimized sequence of vaccine construct was used for *in-silico* insertion into pET30b (+) between XhoI and NdeI restriction sites (Fig. 7).

## DISCUSSION

Kyasanur forest disease is spreading to newer areas. Vaccination has been one of the primary preventive measures. However, the efficacy of the existing vaccine may vary, and it is crucial to boost its effectiveness or develop new vaccines with improved efficacy. A highly immunogenic vaccine is required to fight against this disease and its rapid dispersal to newer areas. The availability of genomic data and immunoinformatics tools aid in development of epitope-based vaccines that can be screened in an efficient manner. Many effective vaccines against infectious diseases designed using reverse vaccinology approaches (that rely on immunoinformatics tools) have been experimentally validated (*Pizza et al., 2000*; *Masignani, Pizza & Moxon, 2019*; *Ismail, Ahmad & Azam, 2020*).

Different research groups have gainfully utilised computational vaccine design approaches for predicting B- and T-cell epitopes to enhance the pace of vaccine development (*Pizza et al., 2000*; *Masignani, Pizza & Moxon, 2019*; *Arumugam & Varamballi, 2021*; *Dey et al., 2023*; *Hafeez et al., 2023*). We have analyzed genome sequences isolated during 1957–2022, and it was noted that there is no evidence of genetic recombination among the isolates.

In this study, the envelope protein of KFDV was chosen to design a multi-epitope vaccine candidate. Previous studies conducted on KFDV demonstrated the suitability of the envelope protein as a candidate for new vaccines and diagnostics (*Dey et al., 2023*). The envelope protein is responsible for the entry of the virus into the host cells by receptor binding and fusion of viral and cellular membranes (*Trapido et al., 1959*). The Ecto-domain-III (EIII) of the envelope protein of flaviviruses evokes neutralizing epitopes (*Liu*

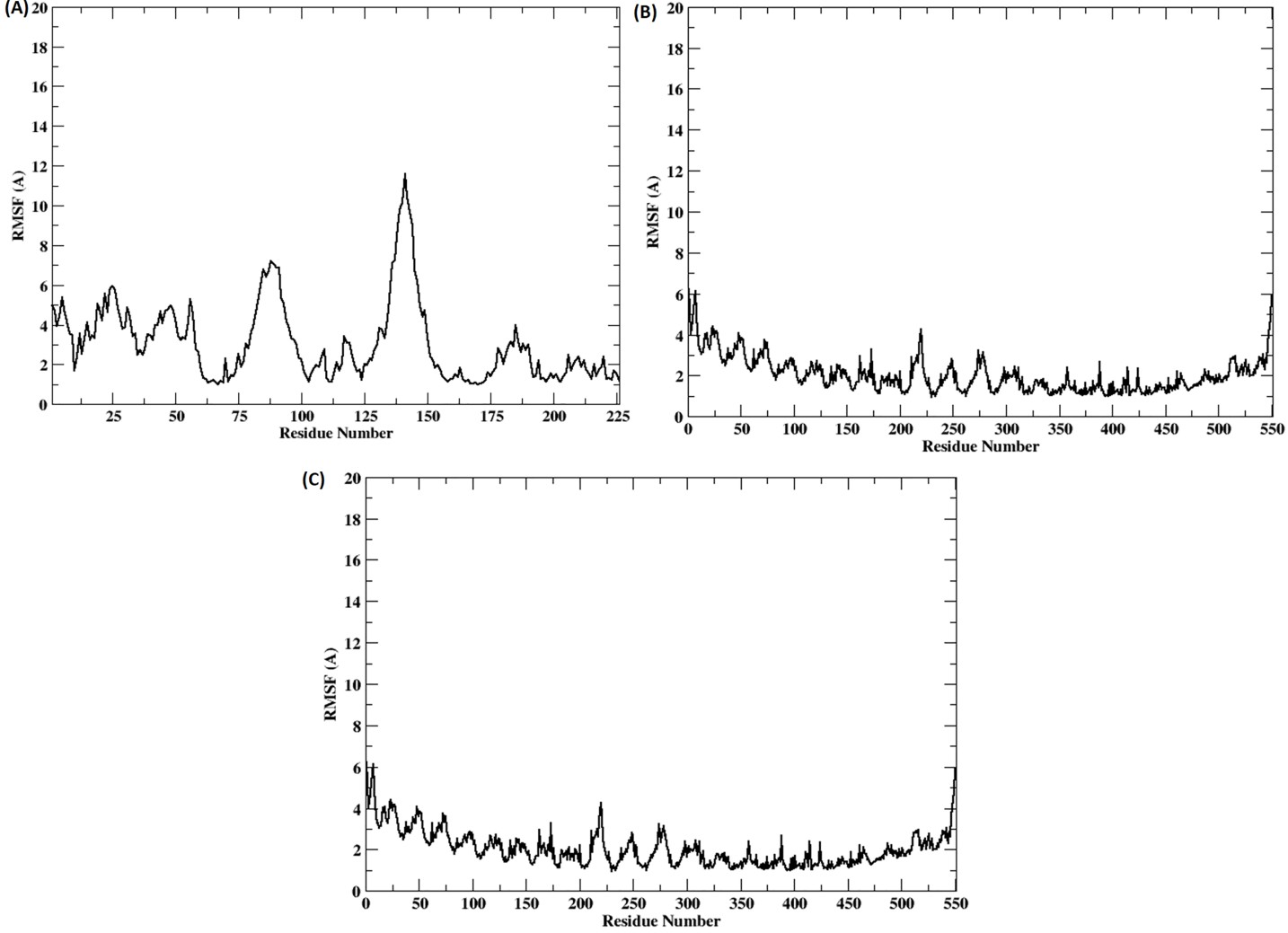

**Figure 5** (A–C) RMSF of the vaccine construct docked with (A) TLR-2 (B) TLR6 (C) TLR2-TLR6 receptor complex.

*et al., 2015*). In earlier studies on immune response, a notable B- and T-cell response was observed in KFDV-infected patients (*Devadiga et al., 2020*). This suggests the need to develop a potential vaccine for KFD that can elicit both humoral and cellular immune responses. Different immunoinformatics tools were used for the prediction and prioritization of a set of B-cell and T-cell (MHC I, and MHC II) epitopes. MHC class I and MHC class II epitopes of envelope protein were predicted using HLA reference alleles. MHC molecules present the peptide epitopes to the T-cell receptors (TCR). MHC I present the peptides to cytotoxic T lymphocytes through the cytosolic pathway and MHC II to the helper T lymphocytes through the endocytic pathway. The non-allergen, non-toxic, IFN-positive epitopes that can stimulate IL-4 and IL-10 cytokines (for MHC-II epitopes) were selected for the final vaccine construct. We designed the multi-epitope vaccine construct by joining the selected predicted epitopes using linkers. There are many *in-silico* multi-epitope vaccine candidates for KFDV and other infectious diseases (*Hafeez et al.,*

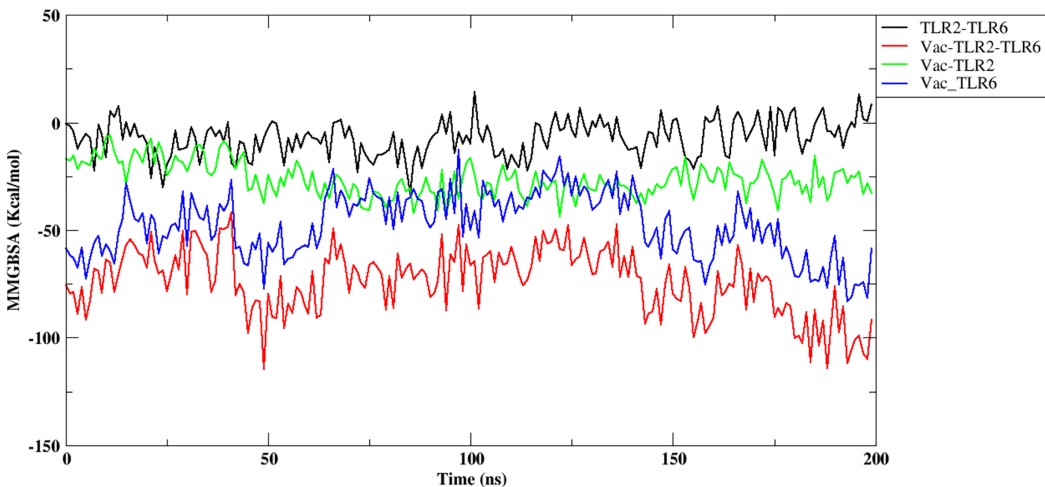

**Figure 6** Free energy of binding between vaccine construct and TLR2-TLR6 complex. TLR2 and TLR6 (black), vaccine construct and TLR2-TLR6 complex (red), vaccine construct and TLR2 (green), vaccine construct and TLR6 (blue).

*2023*; *Ikram et al., 2023*; *Naz et al., 2020*, *2023*). The glycosylation pattern of the epitope has effects on the immunogenicity of the vaccine (*Dowling et al., 2007*; *Singh et al., 2020*). The prioritized epitopes were compared with the key amino acids of the envelope protein. One of the selected epitopes 57-REYCLHAKLANSKVAARCPAM-77 contains the Asp67. The N-glycosylation of the Asp67 of E-protein is responsible for the dengue viral assembly or exit and dendritic cell infection by interacting with DC-SIGN receptors (*Pokidysheva et al., 2006*). In the studies conducted by *Dey et al. (2023)* Asp67 of E-protein in KFDV was predicted to play a role in viral recognition and infection. The ecto-domain-III (EIII) contains the receptor binding region of the envelope protein. The amino acids that form the receptor binding regions are K315, L388, H390, Q391, K395, and F398 (*Dey et al., 2023*). The epitope 384-YVGELSHQW-392 of the vaccine construct includes L388, H390, and Q391 that are likely to enhance the receptor binding ability of the multi-epitope vaccine. Usually, the purified proteins or peptides are less immunogenic, so there is a need to add an adjuvant at the terminal of the peptide to enhance immunogenicity. To increase the immunogenicity of the vaccine construct, β-defensin adjuvant was added (*Kim et al., 2018*) at the N-terminal end along with His tag at the C-terminal end which may aid in purification of the protein in future experimental validation studies. Toll-like receptors (TLR) recognize the microbial components and elicit immune responses by inducing innate immunity followed by adaptive immunity (*Barton, 2007*; *Xagorari & Chlichlia, 2008*; *Zheng et al., 2021*). Many studies have reported the role of TLR2 as a host receptor for the envelope protein to evoke the innate immune response (*Zheng et al., 2021*). TLR2 is the most ubiquitous and is the only TLR that can form heterodimers with more than two other types of TLRs. It forms a heterodimer with TLR6. It interacts with a large number of other non-TLR molecules, thereby increasing its capacity to recognize pathogen-associated molecular patterns (PAMPs) (*Zähringer et al., 2008*). The activation of the TLR2-TLR6 heterodimer initiates a cascade of immune responses that play a dual role in antiviral defense. Depending on the context and extent of activation, this signaling can either inhibit

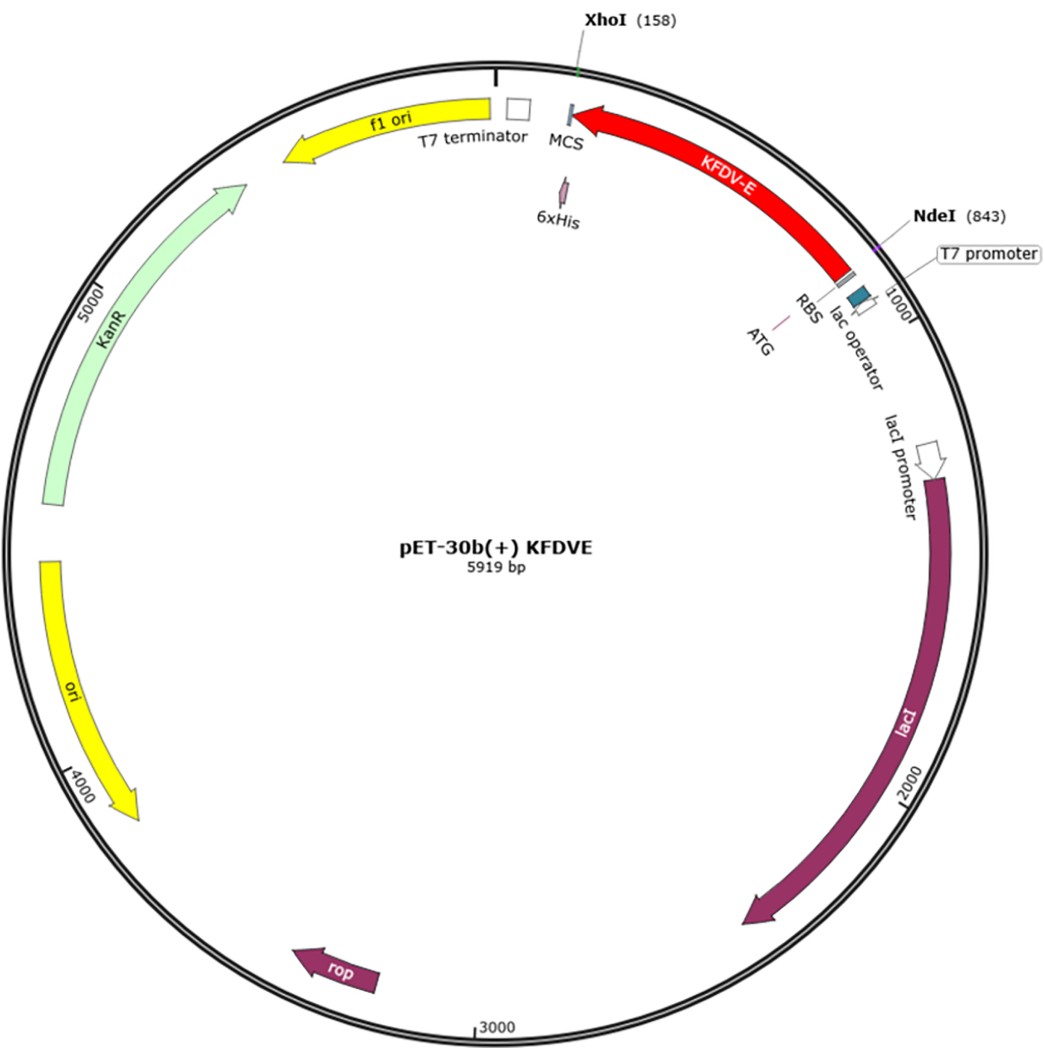

**Figure 7** *In-silico* **cloning of multi-epitope vaccine in pET30b (+).** The red color represents the epitope vaccine sequence inserted between NdeI and XhoI.               

viral replication or promote inflammation and tissue damage. The NS1 protein of dengue virus (DV) has been identified as a critical ligand that activates this receptor complex, leading to an upregulation of pro-inflammatory cytokines, including IL-6 and TNF-alpha (*Chen, Ng & Chu, 2015*). However, evidence from studies conducted by *Modhiran et al. (2017)* suggests that TLR4, rather than the TLR2-TLR6 heterodimer, mediates this activation. Beyond DV, the TLR2-TLR6 pathway also contributes to the innate immune response against respiratory syncytial virus (RSV), indicating its broader relevance in antiviral defense mechanisms (*Murawski et al., 2009*). TLR2 also interacts with molecules like human β-defensin-3, heat shock proteins, and high mobility group box 1 protein and considers them as endogenous ligands (*Funderburg et al., 2007*). We have used the β-defensin as an adjuvant, which can be recognized by TLR2-TLR1 and TLR2-TLR6 heterodimers, increasing the interactive ability of the vaccine construct with the TLR. The heterodimerization increases the range of recognizable motifs by the receptors. In previous

studies of KFDV multi-epitope vaccine, molecular docking was done only with TLR2 (*Arumugam & Varamballi, 2021*; *Hafeez et al., 2023*), which is a major limitation as TLR2 also recognizes the viral proteins in their heterodimeric forms (*Cuevas & Ross, 2014*). Thus, in the present study, the ability of the TLR2 to form a heterodimer with TLR6 was considered, and the vaccine construct was docked with TLR2-TLR6 complex. The docking and simulation study showed that the epitope regions of the vaccine construct formed hydrogen bonds and Van der Waals interactions with both TLR2 and TLR6 and most of the interactions were stable throughout the simulation time. Further, the MM-GBSA results also showed strong free energy of binding of vaccine construct with the TLR2-TLR6 complex. Thus, the study indicates that the vaccine construct forms a stable complex with TLR2-TLR6.

Codon optimization was done to enhance the translation efficiency in *E. coli* (Strain K12). The GC content of 59.43% (30–70%) and CAI 0.95 (0.8–1.0) obtained were in the optimum range for protein expression in the target organism.

### Limitations of the study

It needs to be mentioned that in the present study, only the major antigenic protein, namely envelope protein, was used in the design of multi-epitope vaccine candidate. The inclusion of other structural (capsid and membrane) and non-structural (NS1-NS5) viral proteins would provide a more comprehensive immune response. The placement of predicted B- and T-cell epitopes in the vaccine construct is also known to play a role in eliciting the desired immunogenicity; hence, different arrangements of the epitopes need to be verified. Prediction of MHC-I/II and their binding with epitopes promises to provide a more rigorous criterion for epitope prioritization and hence is proposed to be carried out in future studies. Molecular docking studies of the vaccine construct with other TLRs would also enable a comparative analysis of the binding energies.

## CONCLUSION

The current work describes the comprehensive computational analysis of 73 complete genome sequences of KFDV (1957–2022). The goal is to design of an immunoinformatics-based multi-epitope vaccine against KFDV considering predicted T- and B-cell epitopes and represents a proactive approach to address this emerging infectious disease. Molecular docking and simulations of the vaccine construct with TLR2-TLR6 complex indicate stable binding. Further research, including *in-vitro* and *in-vivo* studies, will be necessary to validate the vaccine's efficacy, safety, and potential for clinical use.

## ACKNOWLEDGEMENTS

The authors gratefully acknowledge the encouragement and support extended by Dr. Sheela Godbole. We also acknowledge the excellent support from Dr. Abhinendra Kumar, Mr. Rajen Lakra, Mr. Prasad Sarkale, Mr. Hitesh Dighe, Mr. Deepak Mali, Ms. Pranita Gawande, Ms. Ujjwala Gaikwad, Ms. Kumari Vaishnavi and Mrs. Pratiksha Vedapathak. The authors acknowledge the computing infrastructure provided by Bioinformatics

Resources and Applications Facility (BRAF), Centre for Development of Advanced Computing, Pune.

### Funding

The grant was provided by the Indian Council of Medical Research, New Delhi, India under the extramural project 'Sustainable laboratory network for monitoring of Viral Hemorrhagic Fever viruses in India and enhancing bio-risk mitigation for High risk group pathogen' with the grant number: VIR/28/2020/ECD-1 dated 10.05.2023. The funders had no role in study design, data collection and analysis, decision to publish, or preparation of the manuscript.

### Grant Disclosures

The following grant information was disclosed by the authors:
Indian Council of Medical Research, New Delhi, India: VIR/28/2020/ECD-1.

### Competing Interests

The authors declare that they have no competing interests.

### Author Contributions

- Sunitha Manjari Kasibhatla conceived and designed the experiments, performed the experiments, analyzed the data, prepared figures and/or tables, authored or reviewed drafts of the article, and approved the final draft.
- Lekshmi Rajan performed the experiments, prepared figures and/or tables, authored or reviewed drafts of the article, and approved the final draft.
- Anita Shete performed the experiments, authored or reviewed drafts of the article, and approved the final draft.
- Vinod Jani performed the experiments, analyzed the data, prepared figures and/or tables, authored or reviewed drafts of the article, and approved the final draft.
- Savita Yadav performed the experiments, analyzed the data, prepared figures and/or tables, authored or reviewed drafts of the article, and approved the final draft.
- Yash Joshi performed the experiments, analyzed the data, prepared figures and/or tables, authored or reviewed drafts of the article, and approved the final draft.
- Rima Sahay performed the experiments, authored or reviewed drafts of the article, and approved the final draft.
- Deepak Y. Patil performed the experiments, authored or reviewed drafts of the article, and approved the final draft.
- Sreelekshmy Mohandas performed the experiments, authored or reviewed drafts of the article, and approved the final draft.
- Triparna Majumdar performed the experiments, authored or reviewed drafts of the article, and approved the final draft.
- Uddhavesh Sonavane performed the experiments, analyzed the data, prepared figures and/or tables, authored or reviewed drafts of the article, and approved the final draft.

- Rajendra Joshi conceived and designed the experiments, performed the experiments, authored or reviewed drafts of the article, and approved the final draft.
- Pragya Yadav conceived and designed the experiments, performed the experiments, authored or reviewed drafts of the article, and approved the final draft.

### DNA Deposition

The following information was supplied regarding the deposition of DNA sequences:

The KFDV sequences are available at GenBank: MG720114, OR162010, MG720091, OR162020, MG720116, OR179916, OR162012, OR162018, MG720114.

### Data Availability

The analysis of molecular dynamics simulations are available in the Supplemental Figures.

The input files and trajectories of molecular simulations of the three replicates of the KFDV vaccine construct are available at Figshare: Kasibhatla, Sunitha; Rajan, Lekshmi S.; Shete-Aich, Anita; Jani, Vinod; Patil, Savita; Joshi, Yash; et al. (2024). KFDV vaccine construct molecular simulations data. Figshare. Journal contribution. https://doi.org/10.6084/m9.figshare.26404432.v1.

### Supplemental Information

Supplemental information for this article can be found online at http://dx.doi.org/10.7717/peerj.18982#supplemental-information.

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
