# Peer review of "Construction of an immunoinformatics-based multi-epitope vaccine candidate targeting Kyasanur forest disease virus"

_PeerJ, doi:10.7717/peerj.18982_

## Round 0.1 · original submission · Major Revisions

In this manuscript, the authors have used computational approaches to design a multi-epitope vaccine candidate for the Kyasanur Forest Disease (KFD) virus. This work is quite interesting. Please see my comments below:

Line 42; Cite references supporting the statement.
Lines 49-55; To make it clear please include text (a few lines) mentioning the structural protein encoded by the single polyprotein mentioned in line 50.
Lines 56-57; "Systematic treatment... approved drug against KFDV"- Please rephrase this sentence to make it clear.
Lines 56-59, 66-67; Cite references to support the statement.
Line 98-99; "Prioritized B- and T-cell epitopes from the envelope protein of KFDV were used for the vaccine design" - Fix the sentence
Lines 109-110; Provide the accession numbers of the data used in this study.
Lines 123-124; Fix the sentence
Line 128; Provide reference for Figtree
Line 131; Include weblink for NetMHCpan EL 4.1 and NetMHCIIpan 3.2. Please provide the link to online tools/web servers throughout the manuscript.
Lines 243-246, 286-290, 292-295, 311-312: move the text to the method section.
Line 278; Please provide a cutoff (threshold) which is used to consider whether the quality of the model is good or bad. And cite the reference to support the statement.
Line 301; Provide the full form of RMSF.
Figure 1: Design section: Replace the arrow between the B-cell epitope and T-cell epitope. At present, the arrow is giving the impression that the T-cell epitopes were predicted from B-cell epitopes.
Too many figures and tables in the main manuscript. Please include some of them in the supplementary material.

Reviewer 1 ·

Basic reporting

Review report is attached

Experimental design

Review report is attached

Validity of the findings

Review report is attached

Additional comments

Review report is attached

Annotated reviews are not available for download in order to protect the identity of reviewers who chose to remain anonymous.

Reviewer 2 ·

Basic reporting

No comment

Experimental design

No explicit research question

Validity of the findings

No comments

Additional comments

Additional computational predictions required: Il-4, IL-10, toxicity and allergenicity

·

Basic reporting

This submission needs revision.

Experimental design

See report

Validity of the findings

See report

Additional comments

1. Abstract: Please mention about the viral proteins wherefrom the epitopes were selected.
2. Introduction: Add recent anti-viral and anti-pathogenic vaccine discoveries through reverse phase vaccinology for translation to therapeutic use.
3. Methods: Section 2.2 can be shortened.
4. Figure: Figure 2 and 3 as well as Figure 4 and 5 can be merged to reduced 4 to 2 Figures. Figure legends need elaboration.
5. Tables: Table 1 can be moved to supplementary Material.
6. Roles of different TLRs in mediating and regulating innate immune response to pathogen and exploitation of the same as vaccine targets should be elaborated
7. A few sentences are unclear and may require rephrasing, a thorough check is required.

---

## Round 0.2 · Minor Revisions

Thank you for submitting the revised manuscript. Other than the minor comments from the reviewer, I have no further comments. Please revise.

Reviewer 1 ·

Basic reporting

Comments attached for reference

Experimental design

Comments attached for reference

Validity of the findings

Comments attached for reference

Additional comments

Comments attached for reference

Annotated reviews are not available for download in order to protect the identity of reviewers who chose to remain anonymous.

---

## Round 0.3 · accepted · Accept

The authors have addressed all the comments. The revised manuscript is ready for publication.